# Automatic Termination for Hyperparameter Optimization

Anastasiia Makarova[1]  Huibin Shen[*2]  Valerio Perrone[2]  Aaron Klein[2]
Jean Baptiste Faddoul[2]  Andreas Krause[1]  Matthias Seeger[2]  Cedric Archambeau[2]

[1]ETH Zürich, [2]Amazon Web Services

**Abstract**  Bayesian optimization (BO) is a widely popular approach for the hyperparameter optimization (HPO) in machine learning. At its core, BO iteratively evaluates promising configurations until a user-defined budget, such as wall-clock time or number of iterations, is exhausted. While the final performance after tuning heavily depends on the provided budget, it is hard to pre-specify an optimal value in advance. In this work, we propose an effective and intuitive termination criterion for BO that automatically stops the procedure if it is sufficiently close to the global optimum. Our key insight is that the discrepancy between the true objective (predictive performance on test data) and the computable target (validation performance) suggests stopping once the suboptimality in optimizing the target is dominated by the statistical estimation error. Across an extensive range of real-world HPO problems and baselines, we show that our termination criterion achieves a better trade-off between the test performance and optimization time. Additionally, we find that overfitting may occur in the context of HPO, which is arguably an overlooked problem in the literature, and show how our termination criterion helps to mitigate this phenomenon on both small and large datasets.

## 1 Introduction

While the performance of machine learning algorithms crucially depends on their hyperparameters, setting them correctly is typically a tedious and expensive task. Hyperparameter optimization (HPO) emerged as a new sub-field in machine learning that tries to automatically determine how to configure a machine learning model. One of the most successful strategies for HPO is Bayesian optimization (BO; Močkus, 1975; Chen et al., 2018; Snoek et al., 2012; Melis et al., 2018) - a powerful framework for sequentially optimizing a costly blackbox objective, that is the predictive performance of a model configured with certain hyperparameters. BO iteratively searches for a better predictive performance via (i) training a probabilistic model on the evaluations of the models performance and (ii) selecting the most promising next hyperparameter candidate.

In practice, the quality of the solution found by BO heavily depends on a pre-defined budget, such as the number of BO iterations or wall-clock time. If this budget is too small, BO might result into hyperparameters of poor predictive performance. If the budget is too large, compute resources will be wasted. The latter can be especially fragile in HPO when one cannot fully reduce the discrepancy between the validation and test errors, thus resulting in *overfitting* as we show in our experiments.

A naive approach suggests terminating BO if the best-found solution remains unchanged for some subsequent BO iterations. Though the idea is sensible, it might be challenging to define a suitable number, since it is a fixed, predetermined choice, that does not take the observed data into account. Another approach is to track the probability of improvement (Lorenz et al., 2016) or the expected improvement (Nguyen et al., 2017), and stop the optimization process once it falls below a given threshold. However, determining this threshold may in practice be less intuitive than setting the number of iterations or the wall-clock time. Instead of stopping BO completely, in McLeod et al. (2018), it is proposed to switch to local optimization when the global regret is smaller than a pre-defined target. This condition can also be used to terminate BO early, but it

---

[*]Correspondence to: Huibin Shen <huibishe@amazon.com>

comes with additional complexity such as identifying a (convex) region for local optimization and again a predefined budget.

Automatically terminating the sequential procedure of BO is a rather under-explored topic, in contrast to the more widely considered orthogonal direction of speeding up HPO via stopping the model training. A seminal idea there is to avoid the computation of low performing hyperparameters, e.g., by learning curves (Swersky et al., 2014), multi-fidelity approach Hyperband (Li et al., 2017), its combination (Klein et al., 2017), and further modifications like Hyperband BOHB (Falkner et al., 2018), asynchronous Hyperband (Li et al., 2020) and its model based version (Klein et al., 2020). The profound distinction of our method with this line of works is in the problem setup: instead of stopping the training, our method aims to terminate the whole HPO process. This orthogonality allows for combination of both ways of to achieve overall larger speed ups.

In this work, we propose a simple and interpretable automatic termination criterion for BO. The criterion consists of two main ingredients: (i) high-probability confidence bound on the regret (i.e., the difference of our current solution to the global optimum) and (ii) the termination threshold. The first already allows a user to specify a desired tolerance that defines how accurate should the final solution be compared to the global optimum. For the case when cross-validation is used, we recommend a threshold based on the statistical properties of the cross-validation estimator. This threshold takes into account the irreducible discrepancy between the actual HPO objective (i.e., performance on new data) and the target function optimized via BO (i.e., the validation error). Our extensive empirical evaluation on a variety of HPO and neural architecture search (NAS) benchmarks suggests that our method is more robust and effective in maintaining the final solution quality than common baselines. We also surface overfitting effects in HPO on both small and large datasets, arguably an overlooked problem, and demonstrate that our termination criterion helps to mitigate it.

## 2 Background

**Bayesian optimization** (BO) refers to a class of methods for gradient-free optimization of an objective $f : \Gamma \to \mathbb{R}$ in an iterative manner. At every step $t$, a learner selects an input $\gamma_t \in \Gamma$ and observes a noisy output

$$y_t \triangleq f(\gamma_t) + \varepsilon_t,$$

where $\varepsilon_t$ is typically assumed to be i.i.d. (sub)-Gaussian noise with some variance $\sigma_\varepsilon^2$. The decision of the next input to evaluate depends on a probabilistic model, used to approximate the objective $f$, and an acquisition function, which determines the decision rule. A popular choice for the probabilistic model is a *Gaussian process (GP)*: $f \sim GP(\mu, \kappa)$ (Rasmussen and Williams, 2006), specified by some mean function $\mu : \Gamma \to \mathbb{R}$ and some kernel $\kappa : \Gamma \times \Gamma \to \mathbb{R}$. As observations $y_{1:t} = [y_1, \ldots, y_t]^\top$ for the selected inputs $G_t = \{\gamma_1, \ldots, \gamma_t\}$ are being collected, they are used to update the posterior belief of the model defined by the posterior mean $\mu_t(\gamma)$ and variance $\sigma_t^2(\gamma)$:

$$\mu_t(\gamma) = \kappa_t(\gamma)^T (K_t + \sigma_\varepsilon^2 I)^{-1} y_{1:t}, \qquad \sigma_t^2(\gamma) = \kappa(\gamma, \gamma) - \kappa_t(\gamma)^\top (K_t + \sigma_\varepsilon^2 I)^{-1} \kappa_t(\gamma), \qquad (1)$$

where $(K_t)_{i,j} = \kappa(\gamma_i, \gamma_j)$ and $\kappa_t(\gamma)^T = [\kappa(\gamma_1, \gamma), \ldots, \kappa(\gamma_t, \gamma)]^T$. The next input to query is determined by an *acquisition function* that aims to trade off exploration and exploitation. Common choices include probability of improvement (Kushner, 1963), entropy search (Hennig and Schuler, 2012) and GP upper-confidence bound (Srinivas et al., 2010) to name a few. The convergence of BO can be quantified via *(simple) regret*:

$$r_t := f(\gamma_t^*) - f(\gamma^*), \qquad (2)$$

where $\gamma^*$ is the global optimizer of $f$ and $\gamma_t^* = \arg\min_{\gamma \in G_t} f(\gamma)$. Specifying adequate tolerance that defines how small the regret should be to terminate BO is of high importance as it determines

both the quality and the cost of the solution. However, this criterion cannot be directly evaluated in practice, as the input $\gamma^*$ and the optimum $f(\gamma^*)$ are not known.

**Hyperparameter optimization** (HPO) is a classical application for BO. Consider a supervised learning problem that requires to train a machine learning model (e.g., a neural network) $\mathcal{M}$ on some feature-response data points $\mathcal{D} = \{(\mathrm{x}_i, \mathrm{y}_i)\}_{i=1}^n$ sampled i.i.d. from some unknown data distribution $P$. The model is obtained by running a training algorithm (e.g., optimizing the weights of the neural network via SGD) on $\mathcal{D}$, both of which depend on *hyperparameters* $\gamma$ (e.g., learning rates used, batch size, etc.). We use the notation $\mathcal{M}_\gamma(\mathrm{x}; \mathcal{D})$ to refer to the prediction that the model produced by $\mathcal{M}$ makes for an input x, when trained with hyperparameters $\gamma$ on data $\mathcal{D}$. Given some loss function $\ell(\cdot, \cdot)$, the *population risk* of the model on unseen data points is given by the expected loss $\mathbb{E}_P[\ell(\mathrm{y}, \mathcal{M}_\gamma(\mathrm{x}, \mathcal{D}))]$. The main objective of HPO is to identify hyperparameters $\gamma$, such that the resulting model minimizes the population risk:

$$f(\gamma) = \mathbb{E}_{(\mathrm{x},\mathrm{y})\sim P}\big[\ell\big(\mathrm{y}, \mathcal{M}_\gamma(\mathrm{x}, \mathcal{D})\big)\big], \qquad\qquad \gamma^* = \arg\min_{\gamma\in\Gamma} f(\gamma). \qquad (3)$$

In practice, however, the population risk cannot be evaluated since $P$ is unknown. Thus, typically, it is estimated on a separate finite validation set $\mathcal{D}_V$ drawn from the same distribution $P$. Practical HPO focuses on minimizing the *empirical estimator* $\hat{f}(\gamma)$ of the expected loss $f(\gamma)$ leading to the optimizer $\gamma_\mathcal{D}^*$:

$$\hat{f}(\gamma) = \frac{1}{|\mathcal{D}_V|}\sum_{(\mathrm{x}_i,\mathrm{y}_i)\in\mathcal{D}_V} \ell\big(\mathrm{y}_i, \mathcal{M}_\gamma(\mathrm{x}_i, \mathcal{D})\big), \qquad\qquad \gamma_\mathcal{D}^* = \arg\min_{\gamma\in\Gamma} \hat{f}(\gamma). \qquad (4)$$

At its core, BO-based HPO sequentially evaluates the empirical estimator $\hat{f}(\gamma_t)$ for promising hyperparameters $\gamma_t$ and terminates after some specified number of BO rounds, reporting the solution $\gamma_t^* = \arg\min_{\gamma\in G_t} \hat{f}(\gamma)$, where $G_t = \{\gamma_1, \ldots, \gamma_t\}$ are the solutions considered so far. We can define the simple regret for the reported solution w.r.t. the validation loss by

$$\hat{r}_t := \hat{f}(\gamma_t^*) - \hat{f}(\gamma_\mathcal{D}^*). \qquad (5)$$

**Inconsistency in the optimization objective.** Importantly, the true HPO objective $f(\gamma)$ in Eq. (3) and the empirical surrogate $\hat{f}(\gamma)$ in Eq. (4) used for tuning by BO generally do not coincide. Therefore, existing BO approaches may yield sub-optimal solutions to the population risk minimization, even if they succeed in globally optimizing $\hat{f}(\gamma)$. This issue, however, is typically neglected in practical HPO as well as a potential overfitting to the validation error. In contrast, we propose a termination condition for BO motivated by the discrepancy in the objectives.

## 3 Termination criterion for Hyperparameter Optimization

This section firstly motivates why early termination of HPO can be beneficial and then addresses the following two questions: (1) How to estimate the unknown simple regret and (2) What threshold of the simple regret can be used to stop HPO.

### 3.1 Motivation for the termination criterion

We start by analysing the effect of optimizing $\hat{f}$ in lieu of $f$. We observe that challenges in optimizing $f$ are both due to the statistical error of the empirical BO objective $\hat{f}(\gamma)$ and the sub-optimality of the BO candidates, encoded in the simple regret $\hat{r}_t$. The key insight of the following proposition is that iteratively reducing $\hat{r}_t$ to 0 may not bring any benefits if the statistical error dominates.

**Proposition 1.** *Consider the expected loss $f$ and its estimator $\hat{f}$ defined, respectively, in Eqs. (3) and (4), and assume the statistical error of the estimator is bounded as $\|\hat{f} - f\|_\infty \leq \epsilon_{st}$ for some $\epsilon_{st} \geq 0$.*

Let $\gamma^*$ and $\gamma_{\mathcal{D}}^*$ be their optimizers: $\gamma^* = \arg\min_{\gamma\in\Gamma} f(\gamma)$ and $\gamma_{\mathcal{D}}^* = \arg\min_{\gamma\in\Gamma} \hat{f}(\gamma)$. Let $\gamma_t^*$ be some candidate solution to $\min_{\gamma\in\Gamma}\hat{f}(\gamma)$ with sub-optimality in function value $\hat{r}_t := \hat{f}(\gamma_t^*) - \hat{f}(\gamma_{\mathcal{D}}^*)$. Then the gap in generalization performance $f(\gamma_t^*) - f(\gamma^*)$ can be bounded as follows:

$$f(\gamma_t^*) - f(\gamma^*) = \underbrace{f(\gamma_t^*) - \hat{f}(\gamma_t^*)}_{\leq \epsilon_{st}} + \underbrace{\hat{f}(\gamma_t^*) - \hat{f}(\gamma_{\mathcal{D}}^*)}_{=\hat{r}_t} + \underbrace{\hat{f}(\gamma_{\mathcal{D}}^*) - \hat{f}(\gamma^*)}_{\leq 0} + \underbrace{\hat{f}(\gamma^*) - f(\gamma^*)}_{\leq \epsilon_{st}} \tag{6}$$

$$\leq 2\epsilon_{st} + \hat{r}_t. \tag{7}$$

Moreover, without further restrictions on $f$, $\hat{f}$, $\gamma_t^*$ and $\gamma^*$, the upper bound is tight.

*Proof:* The equality in Eq. (6) is due to adding and subtracting the same values. The inequality in Eq. (7) results from the following bounds:

(1) $f(\gamma_t^*) - \hat{f}(\gamma_t^*) \leq |f(\gamma_t^*) - \hat{f}(\gamma_t^*)| \leq \max_{\gamma\in\Gamma} |f(\gamma) - \hat{f}(\gamma)| = ||\hat{f} - f||_\infty \leq \epsilon_{st}$,

(2) $\gamma_{\mathcal{D}}^* = \arg\min_{\gamma\in\Gamma} \hat{f}(\gamma) \;\Rightarrow\; \forall \gamma \in \Gamma : \hat{f}(\gamma_{\mathcal{D}}^*) - \hat{f}(\gamma) \leq 0 \;\Rightarrow\; \hat{f}(\gamma_{\mathcal{D}}^*) - \hat{f}(\gamma^*) \leq 0.$ ∎

The proposition bounds the sub-optimality of the target objective $f$ in terms of the statistical error $\epsilon_{st}$ and the simple regret $\hat{r}_t$. This naturally suggests terminating HPO at a candidate $\gamma_t^*$ for which the simple regret $\hat{r}_t$ is of the same magnitude as the statistical error $\epsilon_{st}$, since further reduction in $\hat{r}_t$ may not improve notably the true objective. However, neither of the quantities $\epsilon_{st}$ and $\hat{r}_t$ are known.

Below, we propose a termination criterion that relies on estimates of both quantities. Firstly, we show how to use confidence bounds on $\hat{f}(\gamma)$ to obtain high probability upper bounds on the simple regret $\hat{r}_t$ (Srinivas et al., 2010; Ha et al., 2019). Secondly, we estimate the statistical error $\epsilon_{st}$ in the case of cross-validation (Stone, 1974; Geisser, 1975) where the model performance is defined as an average over several training-validation runs. To this end, we rely on the statistical characteristics (i.e., variance or bias) of such cross-validation-based estimator that are theoretically studied by Nadeau and Bengio (2003) and Bayle et al. (2020).

### 3.2 Building blocks of the termination criterion

**Upper bound for the simple regret $\hat{r}_t$.** The key idea behind bounding $\hat{r}_t$ is that, as long as the GP-based approximation of $\hat{f}(\cdot)$ is well-calibrated, we can use it to construct high-probability confidence bounds for $\hat{f}(\cdot)$. In particular, Srinivas et al. (2010) show that, as long as $\hat{f}$ has a bounded norm in the reproducing kernel Hilbert space (RKHS) associated with the covariance function $\kappa$ used in the GP, $\hat{f}(\gamma)$ is bounded (with high probability) by lower and upper confidence bounds $\text{lcb}_t(\gamma) = \mu_t(\gamma) - \sqrt{\beta_t}\sigma_t(\gamma)$ and $\text{ucb}_t(\gamma) = \mu_t(\gamma) + \sqrt{\beta_t}\sigma_t(\gamma)$. Hereby, $\beta_t$ is a parameter that ensures validity of the confidence bounds (see Appendix A.2.3 for practical discussion and ablation study).

Consequently, we can bound the unknown $\hat{f}(\gamma_t^*)$ and $\hat{f}(\gamma_{\mathcal{D}}^*)$ that define the sub-optimality $\hat{r}_t$:

$$\hat{r}_t = \hat{f}(\gamma_t^*) - \hat{f}(\gamma_{\mathcal{D}}^*) \leq \min_{\gamma\in G_t} \text{ucb}_t(\gamma) - \min_{\gamma\in\Gamma} \text{lcb}_t(\gamma) =: \bar{r}_t, \tag{8}$$

where the inequality for $\hat{f}(\gamma_t^*)$ is due to the definition of the reporting rule $\gamma_t^* = \arg\min_{\gamma\in G_t} \hat{f}(\gamma)$ over the evaluated points $G_t = \{\gamma_1, \ldots, \gamma_t\}$. We illustrate the idea with an example in Fig. 1.

**Termination threshold.** We showed how to control the optimization error via the (computable) regret upper bound $\bar{r}_t$ above. We now explain when to stop BO, i.e., how to choose some threshold $\epsilon_{BO}$ and an iteration $T : \bar{r}_T \leq \epsilon_{BO}$. Following Proposition 1, we suggest setting $\epsilon_{BO}$ to be of similar magnitude as the statistical error $\epsilon_{st}$ of the empirical estimator (since smaller regret $\hat{r}_t$ is not beneficial when $\epsilon_{st}$ dominates). In case of cross-validation being used for HPO, one can estimate this statistical error $\epsilon_{st}$ and we further discuss how it can be done.

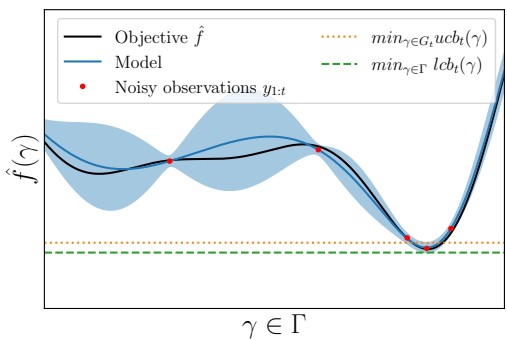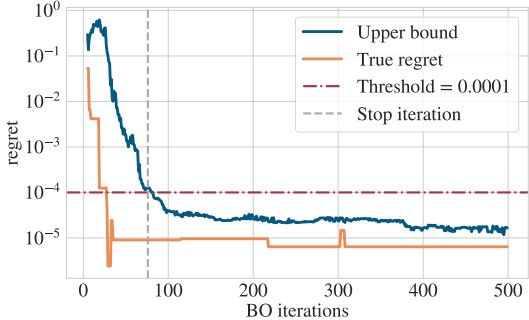

Figure 1: **Left**: Visualization of the upper bound for $\hat{r}_t$. The gap between green and orange lines is the estimate of the upper bound for $\hat{r}_t$. **Right**: Illustration of automated BO termination when tuning MLP on the `naval` dataset from HPO-Bench (Klein and Hutter, 2019) with the BORE optimizer (Tiao et al., 2021).

Cross-validation is the standard approach to compute an estimator $\hat{f}(\gamma)$ of the population risk. The data $\mathcal{D}$ is partitioned into $k$ equal-sized sets $\mathcal{D}_1, \ldots, \mathcal{D}_k$ used for (a) training the model $\mathcal{M}_\gamma(\cdot; \mathcal{D}_{-i})$, where $\mathcal{D}_{-i} = \cup_{j \neq i} \mathcal{D}_i$ (i.e., training on all but the $i$-th fold), and (b) validating $\mathcal{M}_\gamma(\cdot; \mathcal{D}_i)$ on the $i$-th fold of the data. These two steps are repeated in a loop $k$ times, and then the average over $k$ validation results is computed.

The statistical error $\epsilon_{st}$ of an estimator can be characterised in terms of its variance $\text{Var}\hat{f}(\gamma) = \mathbb{E}[(\hat{f}(\gamma) - \mathbb{E}\hat{f}(\gamma))^2]$ and bias $B(\gamma) = \mathbb{E}[\hat{f}(\gamma)] - f(\gamma)$, where the latter can be neglected in case of cross-validation (Bayle et al., 2020). Though the variance $\text{Var}\hat{f}(\gamma)$ of the cross-validation estimate is generally unknown, Nadeau and Bengio (2003) propose an unbiased estimate for it. Specifically, for the sample variance (denoted as $s_{\text{cv}}^2$) of $k$-fold cross-validation, a simple post-correction technique to estimate the variance $\text{Var}\hat{f}(\gamma)$ is

$$\text{Var}\hat{f}(\gamma) \approx \left( \frac{1}{k} + \frac{|\mathcal{D}_i|}{|\mathcal{D}_{-i}|} \right) s_{\text{cv}}^2(\gamma), \tag{9}$$

where $|\mathcal{D}_i|, |\mathcal{D}_{-i}|$ are the set sizes. For example, in the case of 10-fold cross-validation we have $\text{Var}\hat{f}(\gamma) \approx 0.21 s_{\text{cv}}^2(\gamma)$. We are now ready to propose our termination condition in the following.

**Termination condition for BO**. *Consider the setup of Proposition 1 where $\hat{f}(\cdot)$ is a cross-validation-based estimator being iteratively minimized by BO. Let $\gamma_t^*$ the solution reported in round $t$, and $\bar{r}_t$ defined in Eq. (8) be the simple regret bound computed at each iteration $t$. Let the variance $\text{Var}\hat{f}(\gamma_t^*)$ of the estimator $\hat{f}(\cdot)$ be approximated according to Eq. (9). Then, BO is terminated once:*

$$\bar{r}_t < \sqrt{\text{Var}\hat{f}(\gamma_t^*)}. \tag{10}$$

Intuitively, the termination is triggered once the maximum plausible improvement becomes less than the standard deviation of the estimate. This variance-based termination condition adapts to different algorithms or datasets and its computation comes with negligible computational cost on top of cross-validation. The pseudo-code for the criterion is summarised in Algorithm 1. If cross-validation cannot be used or is computationally prohibitive, the user can define the right-hand side of the termination condition. In this case, the upper bound on the left-hand side still has an intuitive interpretation: the user can set the threshold based on their desired solution accuracy. This case is demonstrated in Fig. 1, with an example of automatic termination for tuning an MLP.

## 4 Experiments

The main challenge of any termination criterion for HPO is to balance between reducing runtime and performance degradation. We thus study in experiments how the speed-up gained from different termination criterions affects the final test performance. To this end, we define two new metrics that account for the trade-off between resources saved and performance drop and provide a list of reasonable baselines. The code to reproduce the experimental results is publicly available.[*]

**Baselines**. Since automatic BO terminating is a rather under-explored topic, we consider the following baselines that are, to the best of our knowledge, the only ones directly related to our method:

- Näive convergence test controlled by a parameter $i$ (referred as Conv-$i$): stopping BO if the *best* observed validation metric remains unchanged for $i$ consecutive iterations. It is challenging to define a suitable $i$ suitable across different benchmarks since $i$ is a fix, predetermined choice, that does not take the observed data into account (in contrast to our method that refines the regret estimation). We consider common in practice values $i = \{10, 30, 50\}$ and study other values in Appendix.

- Threshold for Expected improvement (EI; Nguyen et al., 2017): stopping BO once EI drops below a pre-defined threshold. Choosing a threshold crucially depends on a problem at hand, e.g., values studied by the original paper result in too aggressive stopping across a range of our experiments. We thus extend it with a more finer grained grid resulting into $\{10^{-9}, 10^{-13}, 10^{-17}\}$.

- Threshold for Probability of improvement (PI; Lorenz et al., 2016): stopping BO once PI drops below a pre-defined threshold. Similar to EI baseline, we tune the threshold and use $\{10^{-5}, 10^{-9}, 10^{-13}\}$.

**Metrics**. We measure the effectiveness of a termination criterion via two metrics quantifying (i) the change in test error on a held-out dataset and (ii) the time saved. Given a budget of $T$ BO iterations, we compare the test error $y_T$ after $T$ iterations to the test error $y_{es}$ after early stopping is triggered. For each experiment, we compute the *relative test error change* RYC as

$$\text{RYC} = \frac{y_T - y_{es}}{\max(y_T, y_{es})}. \tag{11}$$

This allows us to aggregate the results over different algorithms and datasets, as $\text{RYC} \in [-1, 1]$ and can be interpreted as follows: A positive RYC represents an improvement in the test error when applying early stopping, while a negative RYC indicates the opposite. Similarly, let the total training time for a predefined budget $T$ be $t_T$ and the total training time when early stopping is triggered be $t_{es}$. Then the *relative time change* RTC is defined as

$$\text{RTC} = \frac{t_T - t_{es}}{t_T} \tag{12}$$

indicates a reduction in total training time, $\text{RTC} \in [0, 1]$. While reducing training time is desirable, it should be noted that this can be achieved through any simple stopping criterion (e.g., consider interrupting HPO with a fixed probability after every iteration). In other words, the RTC is not a meaningful metric when decoupled from the RYC and, thus, both need to be considered in tandem.

**Selecting the data for the bound estimate**. Since we are only interested in the upper bound of the simple regret, we conjecture that using only the top performing hyperparameter evaluations may improve the estimation quality. To validate this, we use results of BORE (Tiao et al., 2021) on the `naval` dataset from HPO-Bench (Klein and Hutter, 2019) where we can quantify the true regret (see Section 4.2). We compute the upper bound by Eq. (8) using three options: 100%, top 50% or top 20% of the hyperparameters evaluated so far and measure the distance to the true regret (see Fig. 2).

---

[*]`https://github.com/amazon-research/bo-early-stopping`

From Fig. 2, fitting a surrogate model with all the hyperparameter evaluations poses a challenge for estimating the upper bound of the regret, which is aligned with recent findings on more efficient BO with local probabilistic model, especially for high-dimensional problems (Eriksson et al., 2019). Using the top 20% evaluations gives the best upper bound estimation quality in the median, at the cost of the most under-estimations of the true regret (2553). Our method would stop too early due to the under-estimation, thus negatively impacting the RYC score, as shown in Fig. 2. *As a result, we use the top 50% hyperparameters evaluations for the upper bound estimation throughout this paper.*

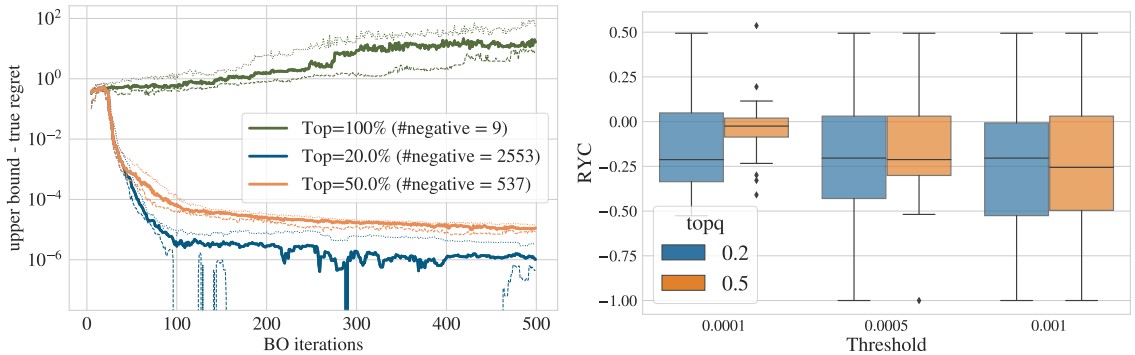

Figure 2: The upper bound estimation quality is affected by the set of hyperparameters evaluations used in the surrogate model training. **Left**: Bound quality for using all, top 50% and top 20% hyperparameter evaluations, measured by the difference between upper bound and true regret. Solid line represents the median over 50 replicates, the dashed is 20'th quantile and dotted line is 80'th quantile. The legend also shows the number of negative differences (the upper bound is smaller than the true regret). **Right**: Box plots of RYC scores when using the top 50% and top 20% hyperparameter evaluations under common thresholds.

## 4.1 BO for hyperparameter tuning with cross-validation

We tune XGBoost (XGB; 9 hyperparameters) and Random Forest (RF; 3 hyperparameters) on 19 small tabular datasets, where we optimize error rate for classification and rooted mean square error for regression, computed via 10-fold cross validation.

**Methods setup.** We use a Matérn 5/2 kernel for the GP and its hyperparameters are found based on a type II maximum likelihood estimation (see Appendix A.1 for more details). The termination is triggered only after the first 20 iterations to ensure a robust fit of GP. We use Eq. (9) to compute our stopping threshold, i.e., $\mathrm{Var}\hat{f}(\gamma) \approx 0.21 s_{\mathrm{cv}}^2(\gamma)$. We additionally use empirical scaling $0.5 s_{\mathrm{cv}}^2(\gamma)$ to study the affect of varying the magnitude.

**Results.** We present the RYC-RTC results aggregated across the datasets in Fig. 3, Table 3 and Figs. 8 and 9 (Appendix A.1). The main take-away from Fig. 3 is as follows: more aggressive early stopping might indeed speed up but it leads to worse test performance, both in terms of average and standard deviation over the datasets. In contrast, the desirable behavior is the trade-off between RYC and RTC, where lower RYC error bars are prioritized over lower RTC error bars. In other words, the methods that adaptively stops BO, i.e., stops when it necessary and does not stop when it is not, is preferable. Fig. 3 indicates that our method successfully maintains a high solution quality across a wide range of scenarios (lower RYC variance) via adapting the termination to the particular propblem (higher variance RTC). The results also show an anticipated RYC-RTC trade-off for $i$ in Conv-$i$ and thresholds in EI and PI baselines, where the solution quality improves as the thresholds increases and consequently the speed-up drops. The average RYC results in Table 3 however are dominated by our method.

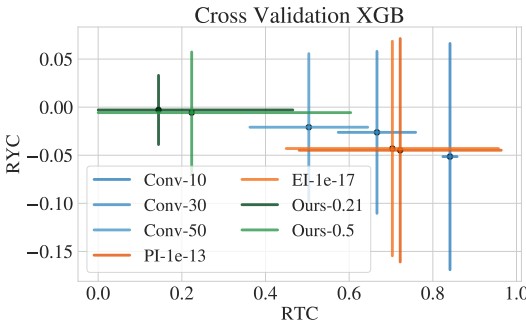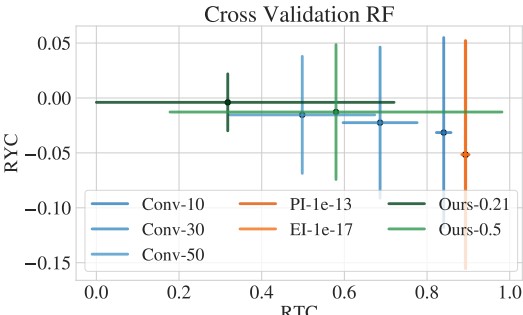

Figure 3: The mean and standard deviation of RYC and RTC scores for the compared automatic termination methods when using cross validation in the hyperparameter evaluation when tuning XGB (left) and RF (right). The mean value is shown as the large dot and the standard deviation is shown as an error bar in both dimensions.

## 4.2 Neural Hyperparameter and Architecture Search

In these experiments we study out termination criterion beyond the BO scope, showing its main advantage of being applicable for any iterative HPO method. In addition, here we also show how to use our method if cross-validation is unavailable. To demonstrate this, we apply it to several state-of-the-art methods: TPE (Bergstra et al., 2011), BORE (Tiao et al., 2021), GP-BO (Snoek et al., 2012) as well as random search (RS) (Bergstra and Bengio, 2012).

**Benchmarks**. We consider two popular tabular benchmark suites: NAS-HPO-Bench (Klein and Hutter, 2019), which mimics the hyperparameter and neural architecture search of multi-layer perceptrons on tabular regression datasets, and NAS-Bench-201 (Dong and Yang, 2020) for neural architecture search on image classification datasets. Notice that for NAS-Bench-201, we used *validation* metrics to compute RYC instead of test metrics, thus, no positive RYC scores are observed. For a detailed description of these benchmarks we refer to the original paper.

**Methods setup**. We consider the following thresholds on the final regret $\{0.0001, 0.001, 0.01\}$, corresponding to a loss of performance of 0.01%, 0.1% and 1%, respectively. Note that, it is easier to set the threshold for our method because it is a threshold on the regret in the metric space that users aim to optimize and it gives more explicit control of this trade-off, thus making it more interpretable. For each method and dataset, we perform 50 independent runs with a different seed.

**Results**. Fig. 4 and Fig. 5 show results for BORE and the rest can be found in the Appendix. While *no method* Pareto dominates the others, our termination criterion shows a similar trend as in Section 4.1 by prioritising accuracy over speed. Users choose the threshold based on their preference regarding the speed-accuracy trade-off, i.e, a higher threshold saves more wall-clock time but potentially leads to a higher drop in performance. We further show the distribution of true regrets at the stopping iteration triggered by our method with the considered thresholds on HPO-Bench in Fig. 5.

From Fig. 5, with a high threshold of 0.01, all the experiments (4 datasets with 50 replicates) are early stopped by our method and 41 (20%) experiments end up with true regret being higher than the threshold. With a low threshold of 0.0001, 112 experiments are stopped and 12 (10.7%) experiments end up with true regret above the threshold. In short, our method achieves 80% to 90% success rate where the true regret is within the user-defined tolerance.

For every method we aggregated the scores over datasets with other HPO optimizers in Fig. 6 (see Appendix A.2). We can see that the speed up of the convergence check baseline is affected very mildly by the optimizers while the RYC scores largely depend on the optimiser: RYC scores with random search are worse than with BORE. In contrast, the RYC scores for our termination criterion are similar across optimisers, especially for smaller thresholds. On the other hand, the speed up

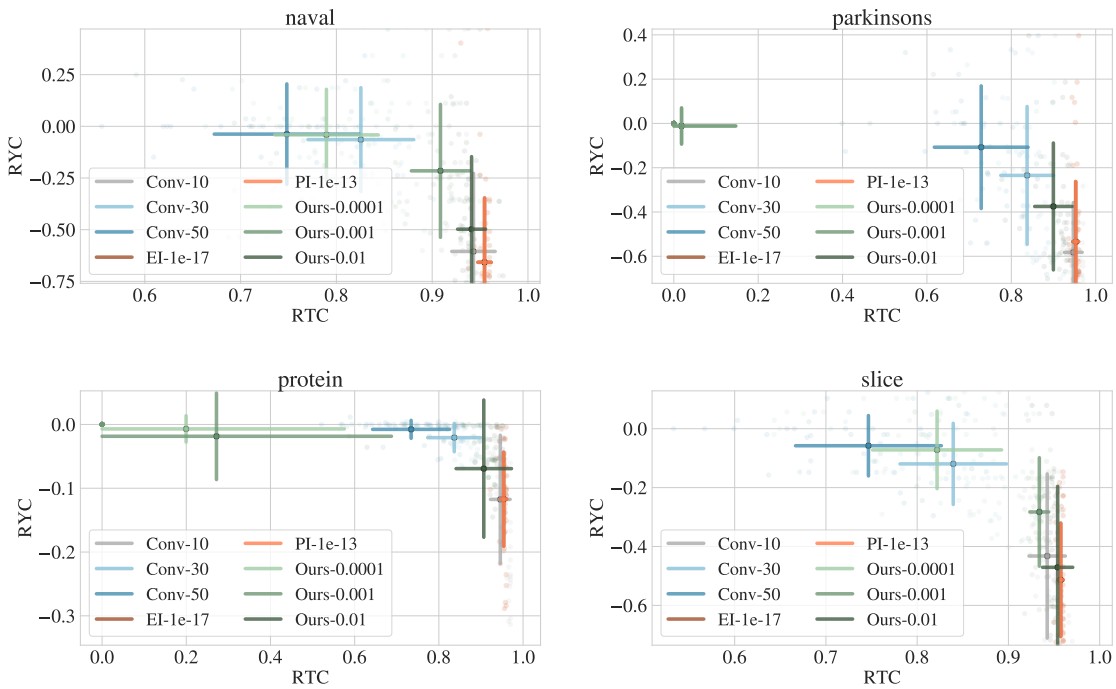

Figure 4: Mean (large dot) and standard deviation (error bar) of RYC and RTC scores for all methods for HPO-Bench datasets.

for a given threshold tends to vary. This can be explained by the difference of the optimizer's performance, for examples random search is not as efficient as BORE, and hence the regret is mostly above the stopping threshold. In summary, while convergence check baselines are by design robust in terms of time saved, our method is more robust in terms of maintaining the solution quality.

### 4.3 Overfitting in BO for Hyperparameter Optimization

Proposition 1 emphasises an important problem of BO-based HPO: while focusing (and minimizing) the validation error, we cannot fully reduce the discrepancy between the validation and test errors. Empirically, we show this might happen when correlation between the test and validation errors is low, thus improvement in validation performance does not lead to the better test results. A particular example of such low correlation in the *small* error region is presented in Fig. 10 (Appendix A.2) when tuning XGB and Random Forest on tst-census dataset.

In our experiments, a positive RYC score is an indicator of overfitting, showing that the test error at the terminated iteration is lower than the test error in the final round. We observe positive RYC scores in both Fig. 3 and Fig. 4, one with cross-validation on small datasets and one with medium-sized datasets. Hence, we would like raise attention to the possible overfitting issue that occurs in HPO for which our method can be used as a plugin to mitigate overfitting.

## 5 Conclusion

Despite the usefulness of hyperparameter optimizations (HPO), setting a budget in advance remains a challenging problem. In this work, we propose an automatic termination criterion that can be plugged into many common HPO methods. The criterion uses an intuitive and interpretable upper bound of simple regret, allowing users explicitly control the accuracy loss. In addition, when cross-validation is used in the evaluations of hyperparameters, we propose to use an analytical threshold rooted from the variance of cross validation results.

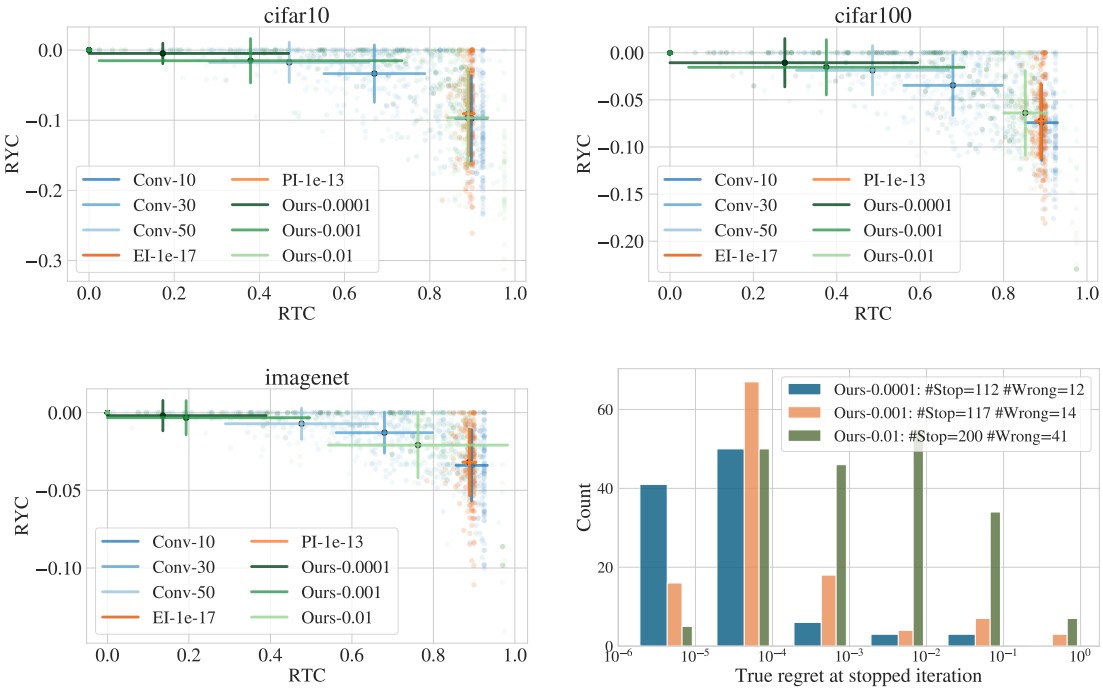

Figure 5: First three figures from the left show the mean and standard deviation of RYC and RTC scores for all methods on NAS-Bench-201. Since *validation metrics* are used in these experiments, no positive RYC scores are observed. The mean value is shown in the big dot and the standard deviation is shown as error bar in both dimensions. The Figure on the right shows a distribution of true regrets at the stopping iteration triggered by our method with different thresholds for HPO-Bench.

The experimental results suggest that our method can be robustly used across many HPO optimizers. Depending on the user-defined thresholds, with 80% to 90% chance, our method achieves true regret within that threshold, saving unnecessary computation and reducing energy consumption. We also observe that overfitting exists in HPO even when cross-validation is used. We hope our work will draw the attention of the HPO community to the practical questions of how to set budget in advance and how to mitigate overfitting when tuning hyperparameters in machine learning.

## 6  Limitations and Broader Impact

Further study of better variance estimate might be beneficial and we leave the modeling and estimation of this variance for the future work. In a broader scope of BO beyond HPO, a similar idea can be used when the point evaluations are not only corrupted by random noise but also adversarial corruptions (resulting again in a discrepancy between the true objective and the computable target).

In a broader context, we highlight that BO can reduce the computational cost required to tune ML models, mitigating the energy consumption and carbon footprint associated with brute force techniques such as random and grid search. The automatic termination criterion we presented in this work can have a positive societal impact by further reducing the cost of tuning ML models.

## 7  Acknowledgements

This research has been gratefully supported by NCCR Automation grant 51NF40 180545. The authors thank the anonymous reviewers of this paper for their helpful feedback.

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

## A Appendix

### A.1 Experiments setting

A.1.1 **BO setting**. We used an internal BO implementation where expected improvement (EI) together with Mat'ern-52 kernel in the GP are used. The hyperparameters of the GP includes output noise, a scalar mean value, bandwidths for every input dimension, 2 input warping parameters and a scalar covariance scale parameter. The closest open-source implementations are GPyOpt using input warped GP [*] or AutoGluon BayesOpt searcher [†]. We maximize type II likelihood to learn the GP hyperparameters in our experiments. We ran 200 iterations sequentially for benchmarks with cross validation, 500 iterations for NAS-HPO-Bench, 200 for NAS-Bench-201.

A.1.2 **Algorithm**. We present the pseudo-code for the termination criterion in Algorithm 1.

---

**Algorithm 1** BO for HPO with cross-validation and automatic termination

---

**Require:** Model $\mathcal{M}_\gamma$ parametrized by $\gamma \in \Gamma$ , data $\{\mathcal{D}_1, \ldots, \mathcal{D}_k\}$ for $k$-fold cross-validation, acquisition function $\alpha(\gamma)$

1: Initialize $y_t^* = +\infty$ and $G_t = \{\}$
2: **for** $t = 1, 2, \ldots$ **do**
3:     Sample $\gamma_t \in \arg\max_{\gamma \in \Gamma} \alpha(\gamma)$
4:     **for** $i = 1, 2, \ldots, k$ **do**
5:         Fit the model $\mathcal{M}_{\gamma_t}(\cdot; \mathcal{D}_{-i})$, where $\mathcal{D}_{-i} = \cup_{j \neq i} \mathcal{D}_i$
6:         Evaluate the fitted model $y_t^i = \frac{1}{|\mathcal{D}_i|} \sum\limits_{\mathrm{x}_i, \mathrm{y}_i \in \mathcal{D}_i} \ell(\mathrm{y}_i, \mathcal{M}_{\gamma_t}(\mathrm{x}_i, \mathcal{D}_{-i}))$
7:     **end for**
8:     Calculate the sample mean $y_t = \frac{1}{k} \sum\limits_k y_t^i$,
9:     **if** $y_t \leq y_t^*$ **then**
10:        Update $y_t^* = y_t$ and the current best $\gamma_t^* = \gamma_t$
11:        Calculate the sample variance $s_{\mathrm{cv}}^2 = \frac{1}{k} \sum_i (y_t - y_t^i)^2$
12:        Calculate the variance estimate $\widehat{\mathrm{Var}} \hat{f}(\gamma_t^*) \approx \left(\frac{1}{k} + \frac{|\mathcal{D}_i|}{|\mathcal{D}_{-i}|}\right) s_{\mathrm{cv}}^2$ from Eq. (9)
13:     **end if**
14:     Update $G_t = G_{t-1} \cup \gamma_t$ and $y_{1:t} = y_{1:t-1} \cup y_t$
15:     Update $\sigma_t, \mu_t$ with Eq. (1)
16:     Calculate upper bound $\bar{r}_t := \min\limits_{\gamma \in G_t} \mathrm{ucb}_t(\gamma) - \min\limits_{\gamma \in \Gamma} \mathrm{lcb}_t(\gamma)$ for simple regret from Eq. (8)
17:     **if** the condition $\bar{r}_t \leq \sqrt{\mathrm{Var} \hat{f}(\gamma_t^*)}$ holds **then**
18:        **terminate BO loop**
19:     **end if**
20: **end for**
21: **Output:** $\gamma_t^*$

---

A.1.3 **Search spaces for cross-validation experiments**. XGBoost (XGB) and RandomForest (RF) are based on scikit-learn implementations and their search spaces are listed in Table 1.

A.1.4 **Datasets in cross validation experiments**. We list the datasets that are used in our experiments, as well as their characteristics and sources in Table 2. For each dataset, we first randomly draw 20% as test set and for the rest, we use 10-fold cross validations for regression datasets and 10-fold stratified cross validation for classification datasets. The actual data splits depend on the seed

---

[*]`https://github.com/SheffieldML/GPyOpt`
[†]`https://github.com/awslabs/autogluon`

Table 1: Search spaces description for each algorithm.

| tasks | hyperparameter | search space | scale |
|---|---|---|---|
| XGBoost | n_estimators | $[2, 2^9]$ | log |
| | learning_rate | $[10^{-6}, 1]$ | log |
| | gamma | $[10^{-6}, 2^6]$ | log |
| | min_child_weight | $[10^{-6}, 2^5]$ | log |
| | max_depth | $[2, 2^5]$ | log |
| | subsample | $[0.5, 1]$ | linear |
| | colsample_bytree | $[0.3, 1]$ | linear |
| | reg_lambda | $[10^{-6}, 2]$ | log |
| | reg_alpha | $[10^{-6}, 2]$ | log |
| RandomForest | n_estimators | $[1, 2^8]$ | log |
| | min_samples_split | $[0.01, 0.5]$ | log |
| | max_depth | $[1, 5]$ | log |

controlled in our experiments. For a given experiment, all the hyperparameters trainings use the same data splits for the whole tuning problem. For the experiments without cross-validation, we use 20% dataset as validation set and the rest as training set.

| dataset | problem_type | n_rows | n_cols | n_classes | source |
|---|---|---|---|---|---|
| openml14 | classification | 1999 | 76 | 10 | openml |
| openml20 | classification | 1999 | 240 | 10 | openml |
| tst-hate-crimes | classification | 2024 | 43 | 63 | data.gov |
| openml-9910 | classification | 3751 | 1776 | 2 | openml |
| farmads | classification | 4142 | 4 | 2 | uci |
| openml-3892 | classification | 4229 | 1617 | 2 | openml |
| sylvine | classification | 5124 | 21 | 2 | openml |
| op100-9952 | classification | 5404 | 5 | 2 | openml |
| openml28 | classification | 5619 | 64 | 10 | openml |
| philippine | classification | 5832 | 309 | 2 | data.gov |
| fabert | classification | 8237 | 801 | 2 | openml |
| openml32 | classification | 10991 | 16 | 10 | openml |
| openml34538 | regression | 1744 | 43 | - | openml |
| tst-census | regression | 2000 | 44 | - | data.gov |
| openml405 | regression | 4449 | 202 | - | openml |
| tmdb-movie-metadata | regression | 4809 | 22 | - | kaggle |
| openml503 | regression | 6573 | 14 | - | openml |
| openml558 | regression | 8191 | 32 | - | openml |
| openml308 | regression | 8191 | 32 | - | openml |

Table 2: Datasets used in our experiments including their characteristics and sources.

## A.2 Detailed results

Fig. 8 demonstrates the results of threshold study EI and PI baselines for cross-validation. Fig. 9 shows results for extended set of parameter $i$ for Conv-$i$ baseline.

We also show the scatter plots of RTC and RYC scores for different automatic termination methods on HPO-Bench-datasets in Fig. 6 and the results on NAS-Bench-201 in Fig. 7.

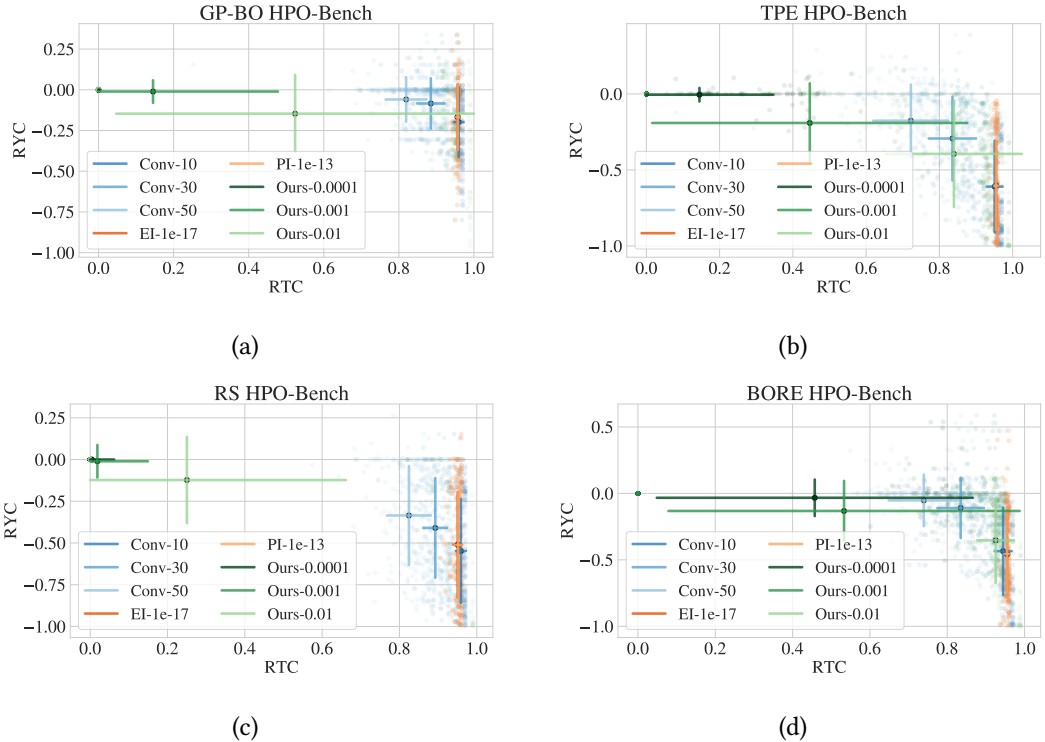

Figure 6: Fig. (a) - (d), the mean and standard deviation of RYC and RTC scores for considered automatic termination methods on HPO-Bench datasets using GP based BO (GP-BO), Random Search (RS), TPE and BORE optimizers. The mean value is shown in the big dot and the standard deviation is shown as error bar in both dimensions.

**A.2.1 Detailed numbers of RYC and RTC scores**. We report detailed RYC scores and RTC scores of different HPO automatic termination methods for the experiments in the main text in Table 3, Table 4 and Table 5.

| | RTC | | RYC | |
| algo | RF | XGB | RF | XGB |
| --- | --- | --- | --- | --- |
| Conv_10 | 0.840 | 0.841 | -0.031 | -0.051 |
| Conv_30 | 0.686 | 0.666 | -0.022 | -0.026 |
| Conv_50 | 0.498 | 0.504 | -0.015 | -0.021 |
| EI_1e-08 | 0.896 | 0.850 | -0.057 | -0.052 |
| EI_1e-12 | 0.895 | 0.779 | -0.055 | -0.047 |
| EI_1e-16 | 0.893 | 0.718 | -0.052 | -0.045 |
| PI_1e-4 | 0.898 | 0.875 | -0.059 | -0.059 |
| PI_1e-08 | 0.895 | 0.814 | -0.055 | -0.052 |
| PI_1e-12 | 0.894 | 0.739 | -0.055 | -0.044 |
| Ours_0.21 | 0.318 | 0.144 | -0.004 | -0.003 |
| Ours_0.5 | 0.580 | 0.224 | -0.013 | -0.006 |

Table 3: RTC and RYC scores for early stopping methods in cross validation benchmarks.

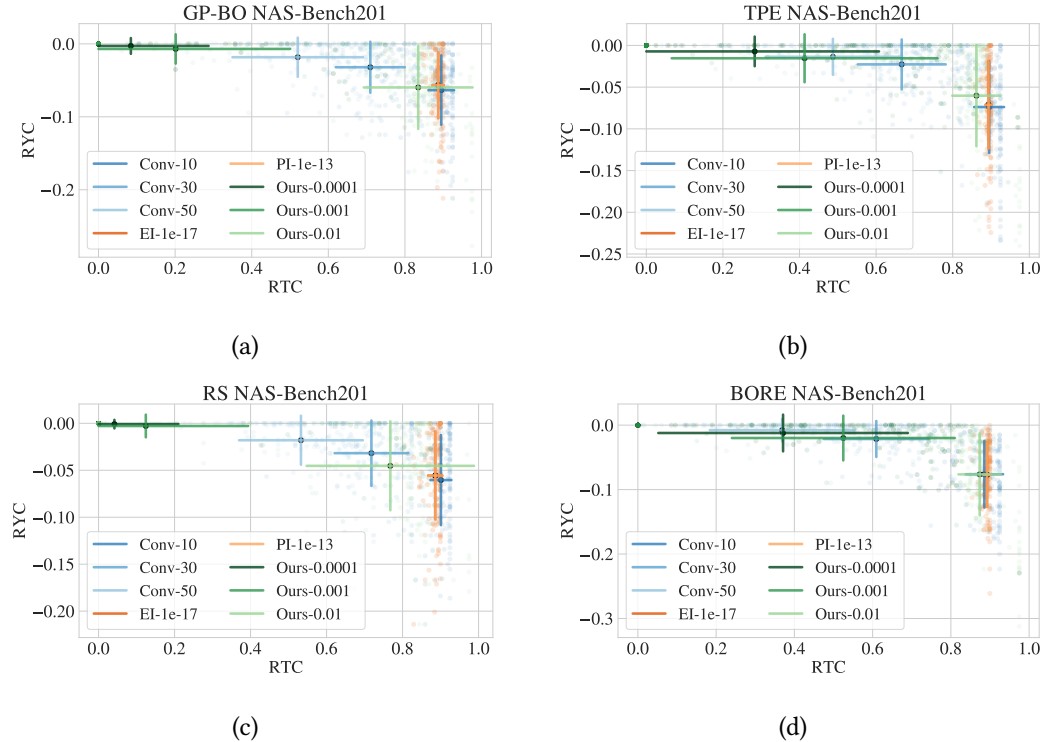

(a)                                         (b)

(c)                                         (d)

Figure 7: Fig. (a) - (d), the mean and standard deviation of RYC and RTC scores for considered automatic termination methods on NAS-Bench-201 datasets using GP based BO (GP-BO), Random Search (RS), TPE and BORE optimizers. The mean value is shown in the big dot and the standard deviation is shown as error bar in both dimensions.

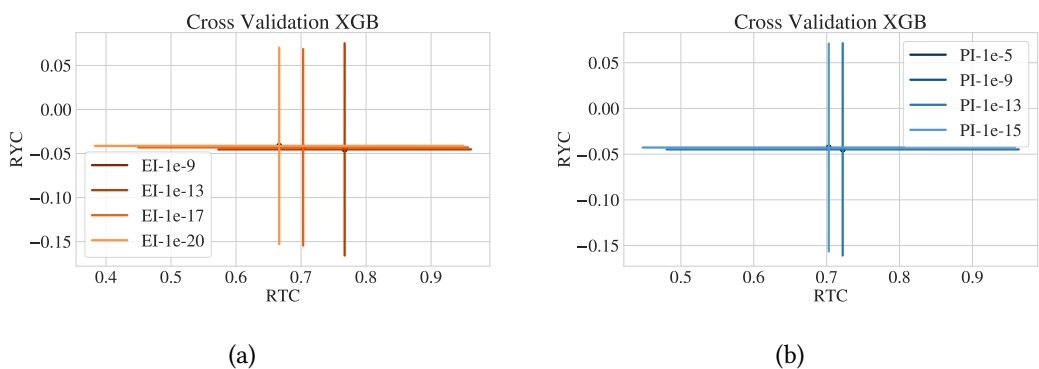

(a)                                         (b)

Figure 8: Detailed threshold study for the EI (left) and PI (right) baselines. Mean and standard deviation of RYC and RTC scores for EI and PI.

**A.2.2 Correlation between validation and test metrics.** In Fig. 10, we show the correlation between validation and test metrics of hyperparameters when tuning XGB and RF on tst-census dataset in Fig. 10.

**A.2.3 The choice of $\beta_t$.** High-probability concentration inequalities (aka confidence bounds) are important to reason about the unknown objective function and are used for theoretically grounded convergence guarantees in some (GP-UCB-based) BO methods (Srinivas et al., 2010; Ha et al., 2019; Kirschner et al., 2020; Makarova et al., 2021). There, $\beta_t$ stands for the parameter that balances

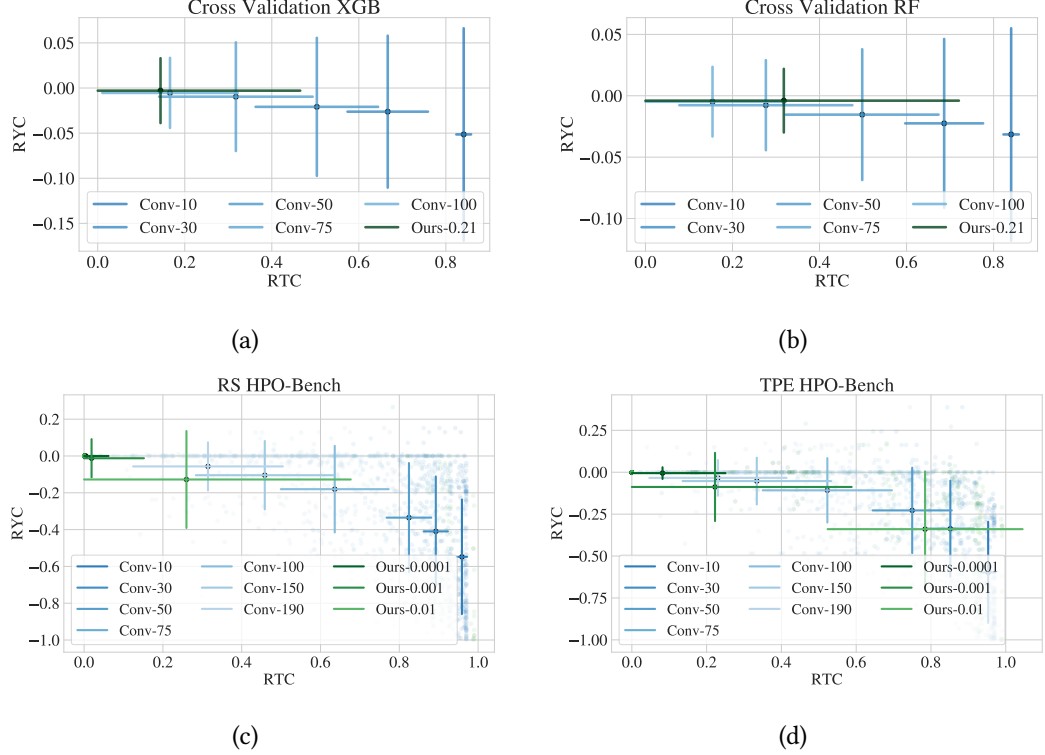

Figure 9: Parameter $i$ study for Conv-$i$ baseline. The mean and standard deviation of RYC and RTC scores for (first line) HPO tuning XGB (a) and RF (b) using cross validation and (second line) for HPO-Bench datasets (c-d). The Figure shows that even though there is an ideal for Conv-$i$, it changes not only across tasks and methods but even across different repetitions of a single method. The latter makes it particularly challenging to define a suitable $i$, since $i$ is a fixed, predetermined choice, that does not take the observed data into account (in contrast to our method that refines the regret estimation as the data is being observed).

| dataset | RTC | | | | RYC | | | |
| | naval | parkinsons | protein | slice | naval | parkinsons | protein | slice |
|---|---|---|---|---|---|---|---|---|
| Conv_10 | 0.943 | 0.947 | 0.946 | 0.942 | -0.605 | -0.582 | -0.117 | -0.432 |
| Conv_30 | 0.826 | 0.837 | 0.837 | 0.840 | -0.064 | -0.235 | -0.021 | -0.119 |
| Conv_50 | 0.748 | 0.729 | 0.734 | 0.747 | -0.038 | -0.107 | -0.008 | -0.058 |
| Ours_0.0001 | 0.790 | 0.018 | 0.198 | 0.822 | -0.041 | -0.012 | -0.005 | -0.072 |
| Ours_0.001 | 0.910 | 0.038 | 0.271 | 0.934 | -0.220 | -0.031 | -0.018 | -0.281 |
| Ours_0.01 | 0.941 | 0.901 | 0.906 | 0.953 | -0.498 | -0.378 | -0.071 | -0.466 |

Table 4: RTC and RYC scores for early stopping methods in HPO-Bench.

between exploration vs. exploitation and ensures the validity of the confidence bounds. The choice of $\beta_t$ is then guided by the assumptions made on the unknown objective, for example, the objective being a sample from a GP or the objective having the bounded norm in RKHS (more agnostic case used in Section 3).

In our experiments, we follow the common practice of scaling down $\beta_t$ which is usually used to improve performance over the (conservative) theoretically grounded values (see e.g., Srinivas et al. (2010); Kirschner et al. (2020); Makarova et al. (2021)). Particularly, throughout this paper, we

| dataset | RTC | | | RYC | | |
| --- | --- | --- | --- | --- | --- | --- |
| | ImageNet | cifar10 | cifar100 | ImageNet | cifar10 | cifar100 |
| Conv_10 | 0.880 | 0.889 | 0.888 | -0.034 | -0.098 | -0.097 |
| Conv_30 | 0.612 | 0.611 | 0.606 | -0.010 | -0.019 | -0.036 |
| Conv_50 | 0.372 | 0.361 | 0.372 | -0.004 | -0.006 | -0.014 |
| Ours_0.0001 | 0.274 | 0.311 | 0.519 | -0.002 | -0.008 | -0.026 |
| Ours_0.001 | 0.377 | 0.622 | 0.582 | -0.005 | -0.023 | -0.033 |
| Ours_0.01 | 0.837 | 0.902 | 0.879 | -0.022 | -0.106 | -0.099 |

Table 5: RTC and RYC scores for early stopping methods in NAS-Bench-201.

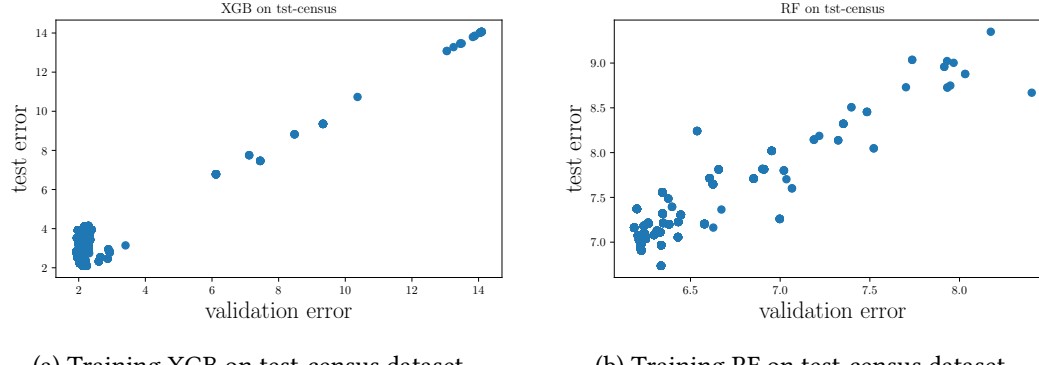

(a) Training XGB on test-census dataset      (b) Training RF on test-census dataset.

Figure 10: We show validation error for training XGB (a) and RF (b) on `tst-census` dataset on the $x$-axis and test error on the $y$-axis. In the *low* error region, the validation metrics are not well correlated with the test metrics.

set $\beta_t = 2\log(|\Gamma|t^2\pi^2/6\delta)$ where $\delta = 0.1$ and $|\Gamma|$ is set to be the number of hyperparameters. We then further scale it down by a factor of 5 as defined in the experiments in Srinivas et al. (2010). We provide an ablation study on the choice of $\beta_t$ in Fig. 11.

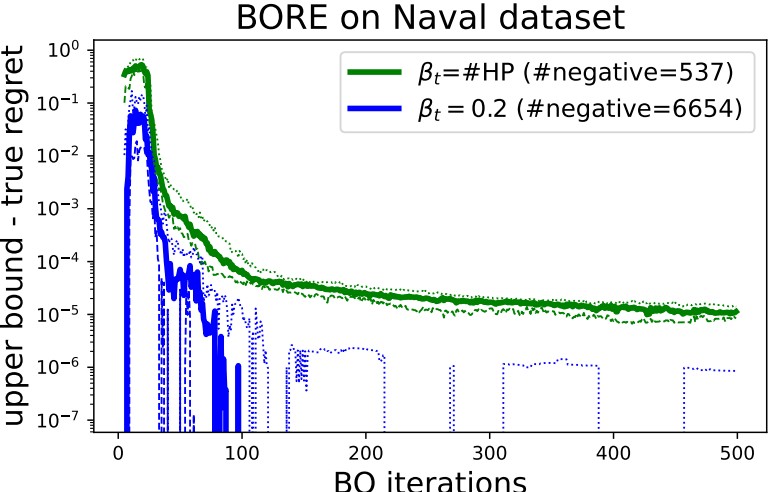

Figure 11: The differences between upper bound and true regret for every BO iterations when using BORE to tune an MLP on the Naval dataset. The number of negative differences (the upper bound is smaller than the true regret) are shown in the legend next to the two options for computing $\beta_t$.

