# OpenReview forum: "Automatic Termination for Hyperparameter Optimization"
_automl.cc/AutoML/2022/Track/Main — AutoML-Conf 2022 (Main Track)_

### Official Review · Reviewer_BzNz · 2022-04-01

**Potential Impact On The Field Of Automl Rating:** 4
**Technical Quality And Correctness:** It is correct.
**Technical Quality And Correctness Rating:** 3
**Clarity Rating:** 3

**Summary Of Contributions:**

This paper proposes an automatic early termination strategy for hyperparameter optimization.  Unlike a general setting of Bayesian optimization, hyperparameter optimization is able to access to a validation dataset as well as a test dataset.  Thus, this paper employs a validation dataset, which is used to measure a validation loss, in determining the automatic termination.  The key insight of this work is that the discrepancy between the true objective and the computable target suggests when we stop the optimization procedure.  The authors provide the empirical results on real-world hyperparameter optimization. Moreover, interestingly the authors find that overfitting may occur in the context of hyperparameter optimization.

**Clarity:**

The writing of this paper is clear.  However, some figures are too small.  Please enlarge them if this work is accepted at AutoML-Conf 2022.

**Ethics Details (Optional):**

I think that this paper does not have any ethical issues.

**Overall Review:**

The authors argue that overfitting may occur in hyperparameter optimization in the last part of the paper, by providing the numerical results.  However, I am a bit convinced of these results and they should be more highlighted in the introduction.

Apart from this, I fully agree with the need for automatic early termination and the strategy proposed in this work is quite promising.  Although I am somewhat worried about using the sample variance of cross-validation in the termination, due to its computation cost and the implication of the sample variance of cross-validation, it is still a reasonable choice.  I would like to listen to thoughts on this concern in the rebuttal.  Please provide thoughts beyond the cases with manual thresholds.

Could I ask the authors how this strategy can be applied in general Bayesian optimization formulation?  If possible, this work is more impactful than the current version which focuses on hyperparameter optimization.

It is a minor issue, but in Figure 1 ``min`` and ``ucb`` should be regular fonts, not italic fonts.

**Potential Impact On The Field Of Automl:**

It is closely related to the AutoML community.  And, in my opinion, this work is potentially impactful in the field of AutoML.

**Reproducibility:**

It seems reproducible by providing an anonymous codebase. I did not run them, though.

**Review Confidence:**

4: You are confident in your assessment, but not absolutely certain. It is unlikely, but not impossible, that you did not understand some parts of the submission or that you are unfamiliar with some pieces of related work.

**Review Rating:**

5: Accept, good paper

**Review Summary:**

By utilizing available resources in hyperparameter optimization, it suggests an interesting and novel technique on automatic termination of hyperparameter optimization.  Therefore, I recommend the decision of this paper as acceptance.

---

### Official Review · Reviewer_5A9q · 2022-04-01

**Potential Impact On The Field Of Automl Rating:** 3
**Technical Quality And Correctness Rating:** 3
**Clarity Rating:** 4

**Summary Of Contributions:**

The paper proposes a new stopping criterion for Bayesian Optimization based on an estimation of the BO model's error and a termination threshold. For estimating the model error, their approach relies on the upper and lower confidence bounds derived in previous work, which is shown to be accurate with high probability. The termination threshold is automatically defined based on cross-validation variance where possible, or defined by a user otherwise. Experiments with multiple benchmarks and baselines are performed to show that the approach achieves a better trade-off between final solution performance and optimization time. As an additional contribution, the paper proposes a discussion on the potential overfitting potential of AutoML solutions.

**Clarity:**

The paper is clearly written and is easy to follow, I commend the authors on this. I have, however, some suggestions I'd like to see adopted for improved clarity in some parts.

* Expand on the initial result in Proposition 1. It shows that a bound can be achieved by a sum of four components. It is clear how those components can be shown to be $\leq 2\epsilon_{st} + r_t$, but not initially clear why those four components can be used to bound the error. The reader is left to analyze by themselves how those four terms came to be in order to establish if it is reasonable, which is possible to be done but would be better if done explicitly by the authors.
* What does "0.21" and "0.5" stand for in the label in Fig 3?
* Figures 4 and 5 are too small, please make them bigger.

Minors:
* Line 59. "Bayesian optimization (BO) refers to methods for gradient-free optimization of an objective [...] in an iterative manner. " is a bit too broad of a definition for BO, there are gradient-free iterative optimizers that are not BO. This can be easily fixed by saying that BO is one (or a class) of such methods.
* Line 189. "T iteration" -> "T iterations"
* Line 260. Mention that Figure 6 is in the Appendix.




**Overall Review:**

The paper proposes a neat new automatic stopping criterion for BO, especially for applications where cross-validation is feasible. The problem tackled is relevant and well-motivated. The paper is clear, well-structured, and the building blocks and final solution are well-presented.

Additionally, there are some interesting results in the experiments section. Namely, the method is capable of achieving similar performance to a non-early stopping scenario, while also saving some compute time, with an easy-to-define (sometimes automatic) stopping criterion. When the threshold is user-defined threshold, the method often achieves the desired accuracy level.

My main concern about this paper lies on the experimental issues presented in the technical quality section. Most notably, it is not fully clear how the baselines were chosen for the experiments and how significant the results are, this makes it hard to judge how relevant the approach really is. At the very least, there needs to be a more thorough discussion on the relevance and significance of the results compared to the baselines. It would also be interesting to see more benchmarks where the CV automatic threshold is used.

At last, the paper lacks a more thorough related works discussion.


**Potential Impact On The Field Of Automl:**

Defining the budget attributed to optimization is a recurrent challenge for practical applications of Bayesian Optimization and AutoML, especially for new users or expensive applications such as those with deep network models or complex data types. The paper proposes to tackle this budgetting issue and, thus, could bring a promising positive impact to the field. This will likely be more appealing to those with expensive applications or developers/maintainers of public or commercial AutoML frameworks, but would still bring positive impact to the field as a whole.

**Reproducibility:**

The authors clearly state the experiment setup and provide an anonymous link that can be used to reproduce the experiments. The reproducibility is well done.

**Review Confidence:**

4: You are confident in your assessment, but not absolutely certain. It is unlikely, but not impossible, that you did not understand some parts of the submission or that you are unfamiliar with some pieces of related work.

**Review Rating:**

5: Accept, good paper

**Review Summary:**

The paper is well-written and the proposed approach is interesting, this paper has potential, but I would like to see some of my concerns with the empirical evaluation addressed before it is accepted. I'm recommending that the paper is not accepted in the current state, but I'm willing to increase the score if the concerns are addressed.

**Technical Quality And Correctness:**

The proposed approach is a well-constructed combination of literature building blocks. It uses the theorems and methodology of Upper/Lower Confidence Bounds to estimate BO's model error and an existing estimation of the statistical error of cross-validation for the termination criterion. It is an interesting and well-done combination of existing concepts into a new context, in an intuitive manner. I enjoyed the idea. I have some concerns for the background and experiments, however.

First, there is no related works section. There is a brief mention to three previous approaches in the introduction, some of which are used as baselines in the experiments, but no discussion on previous works. A quick search on termination criteria for Bayesian Optimization or optimization in general shows several results. I would like to see a more thorough discussion on the existing research on this field. Even if previous works are not directly comparable to this work or are all similar to the three directions cited, it is important to make a thorough discussion on this.


Second, I have a few questions regarding the experiments.
* How were the thresholds for the baselines defined? That is not explained in the paper. Without this reasoning, it is hard to decide whether the baselines are sensible for this comparison. For instance, the baselines in Figure 3 have better RTC, but worse RYC than the proposed method. If the baselines were given a stricter threshold so that they run for longer, what would happen to RTC and RYC?
* The proposed metrics are intuitive to a degree, but also make it a bit difficult to assess exactly how they translate to absolute terms. For instance, the proposed approach achieves ~0.05 better RYC compared to other baselines, is that a significant improvement?
* Why were the EI and PI baselines not used for the results in Section 4.2?
These issues aside, I appreciate that the paper provides an analysis of how close it gets to the user-defined threshold in Fig. 5. It is interesting to see that it achieves the desired accuracy 80 to 90% of the times. I also appreciate the discussion on overfitting.

---

### Official Review · Reviewer_ne5a · 2022-04-08

**Potential Impact On The Field Of Automl Rating:** 3
**Technical Quality And Correctness Rating:** 4
**Clarity Rating:** 3

**Summary Of Contributions:**

This paper investigates an automated approach for stopping hyperparameter optimization. The idea is that when the statistical error in the empirical loss overwhelms the loss itself, then it doesn't make sense to continue optimizing because it becomes impossible to separate signal from noise.

To do this, the paper bounds the true regret in terms of the statistical error and the regret wrt the validation loss. The error can be estimated via cross validation while the validation regret can be bounded by a combination of UCB and LCB (confidence bounds).

For a given regret tolerance, the approach is shown to minimize regret within this tolerance 80-90% of the time, while avoiding spurious evaluations, thus saving computation.

**Clarity:**

The method/theory is very clear, I quite enjoyed reading this part. The experiments are a little less clear, see earlier comments.

**Overall Review:**

Overall I think this is an elegant and principled way to approach the problem of when to stop HPO. The paper was nice to read, especially the method/theory section.

However, practically, I think that the adaptivity of the method is not really captured in the experiments. The Conv-N baselines seem really strong. In particular, I think that increasing N to 75 or 100 may match the proposed approach across most if not all of the benchmark tasks. The paper would be much more compelling if there were HPO benchmarks where the approach worked well with a single threshold across the benchmarks, while the Conv-N baseline required lots of tuning. Otherwise, I am concerned that this may be too involved for people to use generally unless each trial was indeed extremely expensive.

I also think that the cross-validation point still stands. At least in the deep learning world, which is one of the biggest customers of HPO, cross-validation is not used very often because the models/datasets are typically so big and expensive that slightly better risk estimation does not justify the increase in computation.

**Potential Impact On The Field Of Automl:**

Knowing when to terminate HPO is a very important and unsolved problem. I think that the approach presented here is quite principled and very interesting. In particular the adaptation based on the statistical error of different datasets is very interesting and potentially makes the approach quite robust.

However, it also takes some additional tuning (the threshold) and relies on the GP being a good fit to the data for the theory to work out. I worry about the robustness with respect to this parameter and assumption. It looks like, in general, the best performing threshold (in terms of minimizing validation error) saved around 50% of the computation. For HPO this may not justify the complexity and risks of the approach.

Practically, it also relies on cross validation, which I think is much less common than a single held-out validation set, and this could severely impact uptake.

**Reproducibility:**

I believe this work should be easy to reproduce, and code is provided.

**Review Confidence:**

4: You are confident in your assessment, but not absolutely certain. It is unlikely, but not impossible, that you did not understand some parts of the submission or that you are unfamiliar with some pieces of related work.

**Review Rating:**

5: Accept, good paper

**Review Summary:**

Very nice and elegant approach, well written, but perhaps there are some practical limitations that would impede adoption.

**Technical Quality And Correctness:**

The paper is technically correct, the theory is sound, and the proofs are straightforward.

It took me a bit of time to interpret the RYC/RTC plots, but I acknowledge it is a tricky thing to convey both accuracy and time as there is a fundamental tradeoff between these quantities.

Figure 5, far right is very confusing. I was interpreting it as a bar chart, when it is in a histogram. Why is the regret for 200 iterations distributed more highly than the regret for 112/117?

I also think Figure 1 is a bit confusing. Shouldn't the orange line (UCB) be higher since it adds the variance? Right now it seems to intersect at the mean of the function.

---

### Official Review · Reviewer_MMRB · 2022-04-08

**Potential Impact On The Field Of Automl Rating:** 4
**Technical Quality And Correctness Rating:** 3
**Clarity Rating:** 3

**Summary Of Contributions:**

This paper introduces a new approach of early stopping Hyperparameter Optimizer with Bayesian Optimization. Specifically, the authors first estimate the upper bound of the optimization regret and then compare it with a threshold that is computed with the empirical variance of the sample variance of cross-validation. Empirical experiments show that the proposed method is able to early stop at the points that are sufficiently close to the global optima without overfitting the validation loss.

**Clarity:**

I have the following questions for the authors:
1. In Figure 3, what do "ours-0.21" and "ours-0.5" indicate? I can only find the value "0.21" in line 157 under equation 7, then what does "0.5" represent? On the other hand, if "0.5" means "using the top 50% of the hyperparameters evaluated so far" (line 205-206), then how should I explain the value "0.21"?
2. How is the maximal budget (t_T) decided? in Figure 6, 'Conv-50' has an RTC value of 0.9, I guess that t_T should at least allow 500 evaluations?

Minor issues:
1. In Figure 1, the orange line is located on the observation point, shouldn't that be put on the upper confidence bound of that point?
2. Line 183, PI -> EI?
3. Line 260, Figure 6 is presented in the Appendix, please clarify that.
4. fonts in Figures 4 and 5 are too small
5. Algorithm 1, line 15, equation reference is lost


**Overall Review:**

Overall the idea of this paper is quite interesting. The first part of this paper is well established, the author starts by bounding the gap in generalization performance between the candidate. Then the authors show that the regret can be estimated with a surrogate model. Finally, the authors propose their termination condition for BO. All these parts are well established. However, the main drawbacks exist in the experiments and clarification parts (see my above comments), which makes the conclusion of this convincible.

**Potential Impact On The Field Of Automl:**

Overall the idea of this paper is interesting. The proposed method allows the optimizer to stop earlier when it gets closer to the global optimal. Meanwhile,  the early stopping strategy avoids overfitting to the validation loss. Both aspects, i.e., cost-wise and generalization-wise, are important and require further research in the field of AutoML.

**Reproducibility:**

An anonymous GitHub repository is given. This allows the reproducibility of the paper.

**Review Confidence:**

4: You are confident in your assessment, but not absolutely certain. It is unlikely, but not impossible, that you did not understand some parts of the submission or that you are unfamiliar with some pieces of related work.

**Review Rating:**

5: Accept, good paper

**Review Summary:**

Overall the early-stopping idea introduced in this paper is quite interesting. The first part of this paper is well written and well-motivated However, the experiment parts are not clear enough to me. I hope that the authors could clarify some of my doubts.

**Technical Quality And Correctness:**

The theory part of this paper is well established. However, the experiments
1. in line 181-184, the author mention that they set three different thresholds for PI and EI respectively. However, only one threshold for PI and EI is presented in Figure 3, whereas none of the values is in the pre-defined set in lines 181-184
2. In Section 4.2, the authors claim that their criterion is applicable to any iterative HPO methods. However, this argument also works for PI- and EI- based thresholds. Thus, these two baselines should also be reported in Figures 4 and 5
3. The author claims that "We also provide evidence that these baselines are compared to our method, sensitive to the choices of their own hyperparameters". However, only "Conv-i" has different hyperparameters in the results presented in the Figures. The only result I could find is given in Table 3 in the appendix. However, looking at RYC values (whereas the authors emphasised that "The RVC variances of our method are usually smaller than the baseline ones...", line 230) on RF, it seems that the proposed method is more sensitive to its hyperparameters compared to PI and EI based methods.

---

### Official Review · Reviewer_WwZ2 · 2022-04-10

**Potential Impact On The Field Of Automl Rating:** 3
**Technical Quality And Correctness Rating:** 3
**Clarity:** this work is clearly presented.
**Clarity Rating:** 3

**Summary Of Contributions:**

This paper provides a termination method for BO and shows that its works well for a variety of benchmark problems

**Overall Review:**

This work is nice because the derivation is clear and the conclusion is simple.
The weakness lies beside. The conclusion and method are somehow intuitive and a bit simple, in hindsight.

**Potential Impact On The Field Of Automl:**

The proposed method provided a simple, easy to implement, yet effective criterion. It can be applied widely to many HPO problems.

**Reproducibility:**

this experiment is to reproducable with the provided materials.

**Review Confidence:**

3: You are fairly confident in your assessment. It is possible that you did not understand some parts of the submission or that you are unfamiliar with some pieces of related work.

**Review Rating:**

4: Marginally above the acceptance threshold (use sparsely)

**Review Summary:**

I have done some work on BO and I find this paper interesting. If the presented method is solid and can be applied widely as it claims, it is a quite helpful work.

**Technical Quality And Correctness:**

This work provides rigorous analysis and derivations of its approach. The experiment also supports its conclusion.

---

### Meta-Review · Area_Chair_iV5q · 2022-05-09

**Recommendation:** Accept
**Confidence:** 4

**Metareview:**




This paper proposes a new automatic stopping criterion for Bayesian optimization (BO). This methods holds for applications where cross-validation is feasible. It is a problem that is relevant and well-motivated and the paper is well-presented.

The referees had in the very first round of refereeing some questions on the experimental side section and other main question. Yet, the answers of the authors where quite promising and most referees were willing to increase their valuation of the review rating.

In the end the referees all have a point 5 review rating (4 times a point 5 review rating) and one time a 4 review rating. This makes an 4.8 review rating and therefore close to the best possible. Also for the potential impact, the technical quality and the clarity rating they are all between the 3 and the 4 mean rating.

My own reading matches the referees comments. I leave to the Senior Area Chairs the decisions on whether to accept with a highlight or without a highlight the paper, but I would insist in accept the paper in the conference proceedings.

---

### Decision · Program_Chairs · 2022-05-13

Accept